# The Distribution of Tryptophan-Dependent Indole-3-Acetic Acid Synthesis Pathways in Bacteria Unraveled by Large-Scale Genomic Analysis

**DOI:** 10.3390/molecules24071411

**Published:** 2019-04-10

**Authors:** Pengfan Zhang, Tao Jin, Sunil Kumar Sahu, Jin Xu, Qiong Shi, Huan Liu, Yayu Wang

**Affiliations:** 1BGI Education Center, University of Chinese Academy of Sciences, Shenzhen 518083, China; zhangpengfan@genomics.cn (P.Z.); shiqiong@genomics.cn (Q.S.); 2BGI-Shenzhen, Shenzhen 518083, China; jintao@genomics.cn (T.J.); sunilkumarsahu@genomics.cn (S.K.S.); liuhuan@genomics.cn (H.L.); 3State Key Laboratory of Agricultural Genomics, BGI-Shenzhen, Shenzhen 518083, China; 4Citrus Research and Education Center, Department of Microbiology and Cell Science, IFAS, University of Florida, Lake Alfred, FL 33885, USA; joeyxu@ufl.edu; 5Department of Biotechnology and Biomedicine, Technical University of Denmark, Søltofts Plads 2800 Kgs., 2800 Lyngby, Denmark

**Keywords:** indole-3-acetic acid, bacteria, pathways, genomes, metagenomes

## Abstract

Bacterial indole-3-acetic acid (IAA), an effector molecule in microbial physiology, plays an important role in plant growth-promotion. Here, we comprehensively analyzed about 7282 prokaryotic genomes representing diverse bacterial phyla, combined with root-associated metagenomic data to unravel the distribution of tryptophan-dependent IAA synthesis pathways and to quantify the IAA synthesis-related genes in the plant root environments. We found that 82.2% of the analyzed bacterial genomes were potentially capable of synthesizing IAA from tryptophan (Trp) or intermediates. Interestingly, several phylogenetically diverse bacteria showed a preferential tendency to utilize different pathways and tryptamine and indole-3-pyruvate pathways are most prevalent in bacteria. About 45.3% of the studied genomes displayed multiple coexisting pathways, constituting complex IAA synthesis systems. Furthermore, root-associated metagenomic analyses revealed that rhizobacteria mainly synthesize IAA via indole-3-acetamide (IAM) and tryptamine (TMP) pathways and might possess stronger IAA synthesis abilities than bacteria colonizing other environments. The obtained results refurbished our understanding of bacterial IAA synthesis pathways and provided a faster and less labor-intensive alternative to physiological screening based on genome collections. The better understanding of IAA synthesis among bacterial communities could maximize the utilization of bacterial IAA to augment the crop growth and physiological function.

## 1. Introduction

Indole-3-acetic acid (IAA) is a crucial phytohormone in plant development, controlling many important physiological processes, including cell enlargement and division, tissue differentiation, and responses to light and gravity [1]. The ability to synthesize IAA is a well-characterized trait in many plant growth-promoting bacteria [2]. Several experiments have proven that bacterial IAA enhances plant growth [3,4,5,6]. Paradoxically, bacterial IAA is also known as a notorious virulence factor which could induce gall tumors [7,8]. In addition to its involvement in bacteria–plant interactions, IAA also plays an important role in microbial physiology. For instance, IAA acts as a signaling molecule. The addition of IAA represses the *vir* regulon and *chv* genes of *Agrobacterium tumefaciens* [9]. Overproduction of IAA in *Sinorhizobium meliloti RD64* exhibits resistance to various stresses, such as UV and salinity, and also activates the tricarboxylic acid (TCA) cycle [10]. 

IAA is generally synthesized in two ways, the Trp-dependent and the Trp-independent. However, the Trp-independent pathway is still elusive in bacteria [2]. There are four major Trp-dependent IAA synthesis pathways in bacteria, as shown in Figure 1—indole-3-pyruvate (IPA), tryptamine (TPM), indole-3-acetonitrile (IAN), and indole-3-acetamide (IAM). The IAM pathway is the best characterized pathway in both phytopathogens and symbiotic bacteria [11]. It converts tryptophan into IAA in two steps. Tryptophan is first converted to IAM by the enzyme tryptophan-2-monooxygenase (*iaaM*). In the second step, IAM is converted to IAA by an IAM hydrolase (*iaaH*). In the IPA pathway, IAA is initially transformed into IPA by aminotransferases or L-amino-acid oxidases, and then IPA can be converted into IAA by a two-step ipdC pathway in which IPA is decarboxylated to indole-3-acetaldehyde (IAAld) and then IAAld is converted to IAA by dehydrogenases [2], but in plants IAA can be synthesized from the IPA one-step YUCCA pathway (flavin monooxygenase-like enzyme) [12]. For the TPM pathway, Trp is primarily converted into TPM by a decarboxylase and TPM is directly converted to IAAld by an amine oxidase [13]. IAAld is eventually transformed to IAA by dehydrogenases. Trp can be directly converted into IAAld by a Trp side chain monooxygenase enzyme, which is known as the Trp side chain oxidase (TSO) pathway, however, this pathway is only observed in strains from *Pseudomonas fluorescens* [14,15,16], so the TSO pathway is not described here. The IAN pathway is not well studied in bacteria. The first step is to convert Trp into indole-3-acetaldoxime (IAOX) by an oxidoreductase, however, no orthologues have been identified in bacteria thus far. In the second step, IAOX is converted into indole-3-acetonitrile (IAN) by an indoleacetaldoxime dehydratase. Once IAN is formed, it can be converted to IAA from two pathways, one by nitrilases and the other by the NHase/amidase system. NHase links the IAN and IAM pathways by catalyzing IAN into IAM [17]. It has also been confirmed that multiple synthesis pathways can be present in a single bacterium [18]. However, how bacteria regulate these pathways to avoid excessive IAA accumulations and whether these pathways are functionally redundant is still enigmatic. It is reported that nitrilase-based and NHase/amidase system-based synthesis pathways have very different regulatory mechanisms [19]. What is more, different pathways function at different statuses. *Pantoea agglomerans* is a gall-inducing bacterium and possesses IPA and IAM pathways. It is reported that the IPA pathway is related to plant colonization and the IAM pathway is associated with pathogenicity [20].

The bacterial IAA biosynthetic capacity was studied and confirmed primarily by experiments and in silico analysis of individual genomes mainly affiliated to Proteobacteria or Actinobacteria [2]. There is a dearth of information regarding the IAA biosynthetic potential in bacteria across diverse phyla and/or in complex communities. The large-scale genomic analysis could possibly provide insights into whether certain species are potentially capable of IAA synthesis, and how they diversify across different phylogenetic clades. Therefore, in this study, we used thousands of bacterial genomes covering different phyla along with plant’s root metagenomic quantification data and bins to draw an integrated atlas of Trp-dependent IAA synthesis pathways in bacteria. Our findings greatly advanced our understanding of the complex IAA synthesis systems in bacteria and emphasized that IAA is not only a plant growth-promoting trait, but also plays a potentially important role in microbial physiology.

## 2. Results

### 2.1. IAA Biosynthesis is Widely Distributed among Different Bacterial Phyla

We retrieved 7282 complete and draft genomes of microbes across 5 phyla from NCBI, comprising 114 Acidobacteria, 923 Actinobacteria, 281 Bacteroidetes, 1705 Firmicutes, and 4259 Proteobacteria, mostly prevailing in soil and root environments [21]. We surveyed the four major Trp-dependent IAA synthesis pathways in these genomes. In the initial step of each IAA synthesis pathway, Trp was first converted into different intermediates followed by distinct pathways to synthesize IAA. We found that different phyla tended to catalyze Trp into disparate intermediates (*p* value < 0.05, chi-square test), as shown in Figure 2a. In Acidobacteria, Bacteroidetes, and Firmicutes, 65.8% and 48.2%, or 17.4% and 13.5%, or 7.8% and 3.8% of genomes could catalyze Trp into TPM and IAM, respectively. In Actinobacteria, 29.7% and 14.2% could catalyze Trp into IAM and TPM, respectively. In Proteobacteria, 68.5% and 11.9% could catalyze Trp into IPA and TPM. For the IPA pathway, Trp was firstly converted into IPA by an aromatic amino acid or Trp-specific aminotransferases [22] or l-amino acid oxidases [23] in vitro. However, no orthologues of Trp-specific aminotransferases were found in any genomes and only l-amino acid oxidases were observed in Bacteroidetes. In accordance with previous reports, no genomes exhibited the presence of oxidoreductase orthologues that were capable of transforming Trp into IAOX in our data. Overall, these results showed that Trp was likely to be mainly transformed into TPM via Acidobacteria, Bacteroidetes, and Firmicutes, but mainly into IAM by Actinobacteria and into IPA by Proteobacteria.

After examining the de novo IAA synthesis capability of a single bacterial genome from Trp, we found that 50% Acidobacteria, 30.8% Actinobacteria, 15.3% Bacteroidetes, 8.8% Firmicutes, and 19.0% Proteobacteria could potentially de novo synthesize IAA from Trp. Different preferences for biosynthetic pathways were found across phyla (*p* value < 0.05, chi-square test). In detail, 36.0% and 23.7%, 29.0% and 9.4%, 7.1% and 9.3%, 1.8% and 6.5%, or 9.7% and 5.9% of Acidobacteria, Actinobacteria, Bacteroidetes, Firmicutes, and Proteobacteria could synthesize IAA by IAM and TPM pathways, respectively, as shown in Figure 2b and Appendix A. Though 68.5% of Proteobacteria could catalyze Trp into IPA, only 6.6% of those bacteria possessed the intact IPA pathway. Taken together, IAM and TPM were the dominant de novo IAA synthesis pathways in bacteria but varied in phyla.

Although many genomes could not assimilate Trp, some experimental assays verified that some bacteria could synthesize IAA when being supplied with exogenous intermediate metabolites [19], so we scanned how many genomes could potentially synthesize IAA in the presence of key intermediate metabolites (IPA, TPM, IAM, and IAN). Surprisingly, the majority of genomes owned this ability (78.9% of Acidobacteria, 89.4% of Actinobacteria, 63.3% of Bacteroidetes, 78.4% of Firmicutes, and 83.6% of Proteobacteria). IPA could be transformed into IAA by the one-step YUCCA pathway in plants; in our data, we found a small proportion of bacteria contained the homologues of YUCCA, however, the vast majority of bacteria (14.3% of total genomes) used the two-step ipdC pathway, as shown in Figure 2c and Appendix A). IAN could also be transformed into IAA by two distinct pathways, first by one-step nitrilase and the second by two-step NHase/amidase system. Opposite to the IPA pathway, 56.8% of bacterial genomes possess the one-step nitrilase pathway. However, only a minority of bacterial nitrilases identified by in silico analysis were experimentally reported to be capable of converting IAN into IAA [24]. A strong preference for the IAM and TPM pathways were observed in Acidobacteria, and IAM and IPA pathways were dominant in Firmicutes; one-step nitrilase and IAM pathways were primary in Actinobacteria, Bacteroidetes, and Proteobacteria. As aforementioned, only several genomes in Actinobacteria (11 genomes) and Firmicutes (53 genomes) were capable of converting Trp to IPA, but 298 and 379 genomes from these two phyla could potentially synthesize IAA by the two-step ipdC pathway from IPA. These results suggested that the majority of genomes potentially participated in synthesizing IAA from intermediates despite lacking the genes responsible for utilizing Trp, and bacteria from different phyla tended to use incongruent pathways (*p* value < 0.05, chi-square test).

### 2.2. Multiple Distinct Biosynthetic Pathways Coexist in Bacteria

It is already known that multiple IAA synthesis pathways are present in a single bacterium, however, the quantified description of how many IAA synthesis pathways could coexist in a genome is still unexplored. Here, we calculated the number of coexisting disparate IAA synthesis pathways in each genome. The result showed that up to 3 disparate Trp-catalyzing enzymes coexisted in 119 genomes (1 from Acidobacteria, 1 from Actinobacteria, and 117 from Proteobacteria), as shown in Figure 2d and Appendix A. For the majority of genomes, only one Trp-catalyzing enzyme existed. In this study, 32.5% of Acidobacteria contained two disparate Trp-catalyzing enzymes, followed by 10.6% of Actinobacteria. As for the coexistence of various intact IAA synthesis pathways in the individual genome, we observed that five genomes from Proteobacteria possessed three different intact IAA synthesis pathways, both of which simultaneously possess IPA, IAM, and TPM pathways, as shown in Figure 2e, but a plethora of genomes exclusively owned one complete IAM or TPM synthesis pathway. In particular, all the genomes from Firmicutes only contained a single complete pathway. IAM and TPM were the most frequently coexisting intact pathways, as shown in Appendix A. Further detailed investigation of these genomes revealed that 45.3% of genomes possessed unexpectedly complex yet incomplete IAA synthesis pathways, which could potentially produce IAA from intermediate metabolites, as shown in Figure 2f. Moreover, for the genomes which were potentially capable of synthesizing IAA from intermediates, 57.7% of Acidobacteria, 82.6% of Actinobacteria, 43.2% of Bacteroidetes, 28.4% of Firmicutes, and 59.4% of Proteobacteria genomes possessed multiple IAA synthesis pathways (≥2 diverse pathways). A total of 11, 13, 14, 19, and 26 various combinations of coexisting complete or incomplete pathways were found in Acidobacteria, Firmicutes, Bacteroidetes, Actinobacteria, and Proteobacteria, respectively, as shown in Appendix A. Rarefaction analysis showed that the number of distinct combinations in Actinobacteria, Firmicutes, and Proteobacteria had come to saturation due to the great number of genome repertoires, as shown in Appendix A. These results suggest that Firmicutes genomes have the simplest Trp-dependent IAA synthetic systems and a plethora of bacteria could potentially synthesize IAA via multiple incomplete pathways.

### 2.3. Quantification of IAA Synthetic Genes in Root Environments

IAA production is a common feature for plant growth-promoting bacteria, however little is known about how IAA is produced by the complex microbial communities in plant-associated environments. Here we quantified the abundance of different IAA synthesis-related genes in root environments and determined which pathways were dominant in such an ecosystem. We conducted metagenomic sequencing of 27 rhizosphere (RS, the soil microbiota loosely attached to roots) and 19 rhizoplane (RP, the soil microbiota closely attached to roots) samples of foxtail millet (*Setaria italica*) and obtained a non-redundant gene catalogue comprising 67 Mb genes. Even in our deep sequencing data, no orthologues of oxidoreductases were identified that could convert Trp into IAOX. The results also showed that there was no obvious difference in the compositions of IAA synthesis-related genes between rhizosphere and rhizoplane, as shown in Figure 3a,b. Trp was found to be mainly converted into IAM (the abundance relative to Trp-catalyzing enzymes: 54.2% in RP, 54.7% in RS) or TPM (35.9% in RP, 41.4% in RS) and IAA was generally produced by IAM (the abundance relative to IAA-producing enzymes: 37.4% in RP, 43.7% in RS) or IAAld (53.5% in RP, 46.5% in RS) in both rhizosphere and rhizoplane compartments.

Many Trp-catalyzing genes were affiliated to unknown taxa, followed by Proteobacteria and Actinobacteria in both of the two sampling compartments, as shown in Figure 3c and Appendix A. However, the taxonomic compositions of enzymes catalyzing IAA synthesis through intermediate metabolites exhibited distinct results in the two compartments. In rhizosphere, still a large proportion of uncharacterized taxa prevailed, whereas, discrepant to rhizosphere, Proteobacteria and Actinobacteria flourished in rhizoplane, as shown in Figure 3d and Appendix A. Thus, this result suggested that several unreported taxa potentially participated in IAA synthesis and Proteobacteria and Actinobacteria were important agents for IAA synthesis in root environments, indicating that the IAA-producing trait was widely distributed in root-colonizing bacteria. Though there was negligibly slight difference in IAA-synthesis related functional abundance in the two compartments, the taxonomic composition differed a lot. This phenomenon could be attributed to the highly functional redundancy of the IAA-producing trait in root-colonizing bacteria.

### 2.4. The Stronger Capacity of IAA Synthesis for Rhizobacteria

To compare the IAA synthesis abilities between rhizobacteria and retrieved genomes from diverse environments, we binned 563 high-quality metagenomic assembled genomes (≥70% completeness, ≤10% contamination, here we termed these bins as “rhizobacteria”) from our metagenomic data. Phylogenetic annotations showed that 117 bins were affiliated to Proteobacteria and 60 were from Acidobacteria, 49 from Bacteroidetes, 29 from Firmicutes, and 24 from Actinobacteria (for more detailed annotations of all the bins, please refer to Appendix A). We found that 75.0% (422) of these rhizobacteria could convert Trp into intermediate metabolites in the IAA synthesis pathway, while 60.0% and 92.2% of rhizobacteria could synthesize IAA from Trp or intermediate metabolites, respectively. However, the corresponding values in the retrieved genomes were 53.8%, 18.5%, and 82.2%, respectively. Additionally, bins tended to possess multiple distinct IAA synthesis pathways in each genome (the percent of bins and retrieved genomes possessing multiple complete pathways: 38.3% vs. 3.0%; multiple Trp-catalyzing enzymes: 45.6% vs. 10.2%), as shown in Appendix A. These results indicated that rhizobacteria have very high potential to synthesize IAA. Moreover, the ability to synthesize IAA from intermediates IAM and TPM is widely distributed in almost all the rhizobacteria, as shown in Figure 4, which was consistent with the abovementioned results from metagenomic profiling. Intriguingly, 59 out of 60 rhizobacteria from Acidobacteria could transform Trp into IPA but only 2 out of 114 retrieved Acidobacteria owned this capacity. The *iaaM* genes were lost in all the rhizobacteria from Acidobacteria and were only observed in Proteobacteria and Planctomycetes, as shown in Figure 4. Rhizobacteria from Firmicutes were absent in of all the genes participating in Trp transformations, but 14.7% of retrieved Firmicutes possessed these genes. One-step IPA-YUCCA pathway was exclusively observed in one Bacteroidetes, three Firmicutes, and eight Gemmatimonadetes rhizobacterial genomes. These results highlighted that the patterns of IAA synthesis pathways in rhizobacteria were different from bacteria within the same phyla but isolated from other environments, which might be selected by the specific secretions from plant roots.

IAA synthesis-related genes are supposed to locate either on chromosomes or plasmids. Those genes which are located in plasmids usually represent higher levels of transcription, as plasmids frequently occur with high copy numbers in bacterial genomes and have the opportunity to transfer and insert into the plant chromosomes [25]. To identify the location of related genes in each rhizobacterial genome, we predicted the plasmid scaffolds by Plasflow [26] and mapped those genes in the predicted plasmids. We found that the majority of genes were located on chromosomes, however one clade (leaves from bins_290 to bins_328 in Figure 4) from Proteobacteria showed the highest proportions of genes located on plasmids. Rhizobacteria affiliated to Proteobacteria and Acidobacteria tended to locate IAA synthesis-related genes on plasmids compared with rhizobacteria from other phyla. The two phyla showed a close phylogenetic relationship, so this tendency might arise with the bacterial evolution and was possibly maintained by the vertical inheritance.

## 3. Discussion

The access to tens of thousands of genome data facilitates the understanding of the genetic foundations of numerous phenotypes. In this study, we conducted a large-scale genomic analysis to uncover the Trp-dependent IAA synthesis pathways in bacteria and quantified related genes in the root environments by metagenomics. Bacterial IAA is reported to mediate microbial physiology, such as overcoming stress, behaving as a signaling molecule for different cellular processes, and serving as a carbon and nitrogen source [27,28,29]. We found that 18.5% or 82.2% of bacteria from diverse environments could potentially synthesize IAA from Trp or intermediates, respectively, so this phenomenon emphasized that IAA probably has a profound impact on shaping microbial physiology. We speculate that either the required intermediates are rich in their habitat or the transfer of intermediates among individuals in communities might lead to the plethora of bacterial genomes which can synthesize IAA from the intermediates. Similarly, a recent study on cobamide biosynthesis in bacteria also revealed that many bacteria only contained incomplete pathways and the authors experimentally verified that bacteria could truly synthesize cobamide from intermediates [30]. Intriguingly, oxidoreductase orthologues that catalyze Trp into IAOX does not show any hit, neither in the individual genomes nor in the deeply sequenced metagenomes, implying that the sequences are too divergent or the strategy for producing IAOX in microbes is quite different from plants. The IPA pathway is the sole pathway used to study the regulation of IAA synthetic pathways, as IPA is deemed to be the most common pathway in bacteria [2,17]. Different from previous knowledge in our data, we found bacteria from different phyla favored different pathways, but IAM and TPM showed dominance. In the previous studies, the identification of the IPA pathway in bacteria were mainly confirmed by the detection of aminotransferases or ipdC, and most of those scanned bacteria were from Proteobacteria [2]. Consistent with our results, most bacteria from Proteobacteria did contain aminotransferases, but many of them lacked the enzymes required in the subsequent steps of IPA pathway. Hence, mere identification of the genes essential for the IPA pathway does not imply that bacteria could synthesize IAA via this pathway, as multiple pathways could also coexist in a single genome. Therefore, we also characterized the coexisting pathways in each genome. Intriguingly, there were up to three distinct yet intact pathways or four various incomplete pathways that were found to coexist in several genomes. Different pathways were linked to a difference in regulation modes [20], which could help to maintain continuous IAA synthesis under diverse environments.

IAA plays a profound role in plant–microbe interactions. Both beneficial and pathogenic bacteria are able to produce IAA. IAA synthesized by plant growth-promoting bacteria isolated from root environments could participate in nodule formation and increase host root surfaces by promoting lateral roots’ growth [31,32]. To the best of our knowledge, this is the first time to quantify the IAA biosynthesis related genes in root environments. Though, both IAAld and IAM could be produced by two pathways respectively, the high abundance of decarboxylases and iaaM in metagenomic data might suggest that TPM and IAM were the major IAA synthesis pathways in root environments, besides, Proteobacteria and Actinobacteria were essential performers for the IAA synthesis. We binned 563 high-quality rhizobacterial genomes from our metagenomic data. Pioneer works screened the auxin productivity in 1151 plant-associated bacterial strains and verified that 229 strains produced auxin from Trp [33]. However, much higher potential was found in our metagenomic bins, of which 60.0% could potentially de novo synthesize IAA from Trp. This deviation might be caused by erroneous clustering in bins or the inappropriate conditions for detecting IAA synthesis in culture media. When compared with the retrieved genomes from diverse environments, we found that rhizobacteria likely showed more effective IAA synthetic abilities. Having the powerful ability to synthesize IAA must give rhizobacteria a selective superiority in root environments. It has speculated that microbial IAA could promote plants’ susceptibility to both pathogen and plant growth-promoting bacteria [34]. Microbial IAA is assumed to alter auxin homeostasis in plants and the outcomes of this interaction are determined by the endogenous IAA level in plants [35]. Root exudates are rich in carbohydrates and IAA could facilitate microbes to uptake carbohydrates by increasing the expression of ATP-binding cassette transporters and tripartite ATP-independent periplasmic transporters [28]. Microbes also can obtain Trp from root exudates [36] and this might explain why rhizobacteria have the stronger capacity to de novo synthesize IAA from Trp. In addition, IAA could facilitate the expressions of the type VI secretion systems and effectors of type III secretion system which participate in plant–microbe and microbe–microbe interactions in root environments [28,37]. Furthermore, IAA is engaged in inducing antibiotic production which could be equipped as a weapon against other microbes competing for the same resources in root environments or prevent host plants from pathogens [38]. Taken together, microbial IAA could contribute to the successful colonization of microbes in root environments.

## 4. Materials and Methods

### 4.1. Genome Retrieval and Analysis

We searched and downloaded the complete and draft genomes across five phyla from NCBI, including Acidobacteria, Actinobacteria, Bacteroidetes, Firmicutes, and Proteobacteria. Totally, 7282 genomes were retrieved. The downloading ftp for the analyzed genomes can be found in the Appendix A, as shown in Appendix A. We re-predicted the genes in those genomes by Prodigal [39] and then mapped the genes against the KEGG database (Ver. 81, Kanehisa Laboratories, Kyoto, Japan) [40] by Diamond [41] to search for the Trp-dependent IAA synthesis-related genes (map00380, the KEGG orthologies (KOs) for each pathway are listed in Appendix A). The alignment results were filtered using E-value < 1e-5 and alignment score ≥ 60. Only the best hits were retained after filtration. To evaluate the accuracy of functional annotation, all the genomes from Acidobacteria and Bacteroidetes were aligned against the COG database by Blastp [42] and only the KOs with COG tags were considered. The results showed 80.6% of the KEGG-identified genes had the same annotations with the COG-identified genes. Though the COG database identified more IAA synthesis-related genes than the KEGG database did, to decrease the ambiguous alignments, only the annotations from the KEGG database were retained.

### 4.2. Sample Collection for Metagenomic Sequencing

Foxtail millet seeds were sowed in the field in Yangling (Shanxi, China). For each cultivar, the root samples from three individuals were harvested during the ripening stage and separated into rhizosphere and rhizoplane samples using an established protocol as our previous study [21]. All the samples were placed in 15 mL clean microcentrifuge tubes and transferred back to the lab under −20 °C. Rhizosphere samples were collected by scraping the soil loosely attached to the root surface and rhizoplane samples were collected from the tightly adhering soil that was washed off by vortexing for 30 min using the mixture of 1× phosphate buffer saline (PBS) and adsorbent Silwet-77. The DNA extraction procedure was done according to the protocols of the MoBio PowerSoil toolkit (QIAGEN, Inc, New York, NY, USA). Unfortunately, several samples failed to yield good quality DNA from rhizosphere or rhizoplanes. However, we managed to obtain 27 DNA samples from rhizosphere and 19 DNA samples from rhizoplanes.

### 4.3. Metagenomic Sequencing and Processing

Four samples were pooled together in equal amounts and libraries were constructed to be sequenced on the BGISEQ500 platform (BGI, Shenzhen, China). The raw sequencing reads have been deposited in the NCBI BioProject database under the accession number PRJNA450295 and are also accessible at the CNGBdb (China National GeneBank DataBase) under the project CNP0000240. Raw reads were filtered by SOAPnuke (v1.5.3, BGI, Shenzhen, China) [43] with parameter settings as “filterMeta -Q 2 -S -L 15 -N 3 -P 0.5 -q 20 -l 60 -R 0.5 -5 0”. Then the clean reads were aligned against the foxtail millet genome (v2.3, BGI, Shenzhen, China, http://foxtailmillet.genomics.org.cn/page/species/download.jsp) [44] to remove host contaminations using Bowtie2 [45]. Assembled contigs with lengths less than 500 bp were excluded from the subsequent analyses. Genes were predicted over the contigs assembled from each test sample by Prodigal [46] under the “Meta” mode and only genes with a length greater than or equal to 150 bp were retained. All of the nucleic acid sequences of genes were merged and input into CD-HIT [47] with 0.9 coverage and 0.95 identity to generate the non-redundant gene catalog.

Functional annotations were implemented by mapping non-redundant genes against KEGG (Ver. 81, Kanehisa Laboratories, Kyoto, Japan) [48] by Diamond, and filtering criteria were the same as the genome annotation step. Taxonomic annotations were executed by mapping genes to the NCBI “nr” database (release date: 20160219) with Diamond, and alignments were filtered with coverage ≥ 0.8 and identity ≥ 0.65. Taxonomic information was obtained using an in-house lowest common ancestor (LCA) pipeline [49].

The clean reads from each sample sequenced on the BGISEQ500 platform were aligned against the gene catalog by Bowtie2 with parameters “--sensitive --dpad 0 --gbar 99,999,999 --mp 1,1 --np 1 -I 100 -X 500 --score-min L,0,-0.1 -k 50--no-discordant--no-unal --no-sq --no-head”. The abundance of each gene in each sample was calculated as the number of reads mapped to the gene divided by the length of the gene. The KO profile was generated by summing up the abundance of genes that affiliated to the same KO. The taxonomy profile at the phylum or class level was generated by summing up the abundance of genes affiliated to the same phylum or class.

### 4.4. Metagenomic Binning

Metagenomic assembled genomes were generated by the following steps: (1) All of the 46 clean metagenomic data were assembled by Megahit [50]. A total of 8 samples were co-assembled while the remaining 38 samples were assembled separately. Finally, 39 assemblies were generated in total. (2) Only the scaffolds with length ≥ 2000 bp were retained for metagenomic binning. The clean metagenomic reads from all the 46 samples were mapped to each assembly by Bowtie2 [45], resulting in 46 × 39 bam files. (3) Scaffolds from each assembly and the corresponding bam files were used as input files of Metabat2 [51]. (4) The bins from each assembly were evaluated for completeness and contaminations by CheckM [52]. Only those bins with completeness ≥ 70% and contamination ≤ 10% were retained for the following analysis. (5) To remove the redundancy of bins from different assemblies, we calculated the gANI by ANIcalculator [53]. Bins were clustered based on the gANI ≥ 99.9% and alignment fraction (AF) ≥ 90%. However, no bins could be clustered due to the low AF. Each bin was an input into Plasflow [26] to predict the plasmid sequences.

### 4.5. Phylogenetic Annotations

Retained bins were searched for the 16 single-copy marker genes as previously described [54] by HMMsearch (EMBL-EBI, Cambridge, UK) [55]. Bins with ≥ 8 marker genes and all single-copy genes were kept for the phylogenetic analysis. Also, prokaryotic genomes marked as “Reference” or “Representative” in Refseq [56] were downloaded and searched for these marker genes. The protein sequences of each marker gene were multiple aligned by MUSCLE [57] and the columns containing more than 70% of gaps were removed. The concatenated alignments were used to generate the phylogenetic trees by Fastree (Lawrence Berkeley National Laboratory, Berkeley, CA, USA) [58] and visualized using iTOL [59]. The taxonomic annotations of bins were assigned by Tax2tree [60].

## 5. Conclusions

In summary, our work provided a comprehensive atlas of Trp-dependent IAA synthesis pathways across different bacterial phyla. However, experimental assays are needed to verify these conclusions in future. A full understanding of how microbes synthesize IAA may enable the maximization of the beneficial effects of plant growth-promoting bacteria and reduction of the virulence of IAA-producing phytopathogens. Finally, the identification of a high percentage of microbes that could potentially synthesize IAA further calls for in-depth research to explore the role of IAA in mediating microbial physiology.

## Figures and Tables

**Figure 1 molecules-24-01411-f001:**
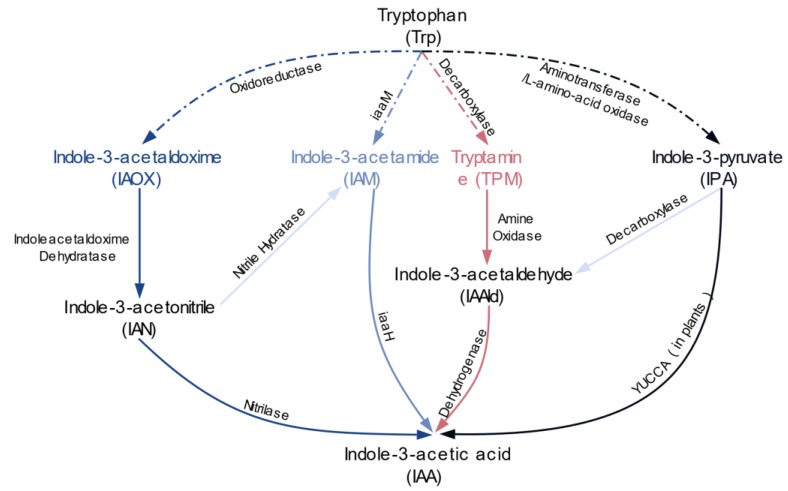
The major microbial Trp-dependent IAA synthesis pathways. Different pathways are depicted with different colors. In our study, we decompose each pathway into three parts: the first part represents the first step of IAA synthesis or the Trp-catalyzing step (dashed lines in the figure); the second part represents synthesizing IAA from Trp or the de novo IAA synthesis; the third part represents synthesizing IAA from intermediates (IPA, IAM, TPM, IAN).

**Figure 2 molecules-24-01411-f002:**
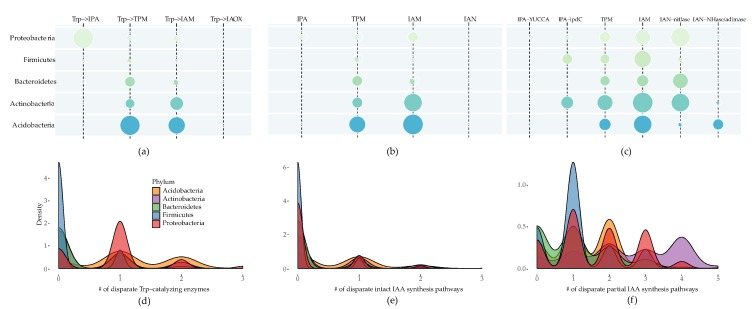
The distribution of IAA synthesis pathways across distinct phylum. The size of bubbles in each bubble plot represents the percentages of detected genomes in each phylum. (**a**) The number of genomes possessing the Trp-catalyzing enzymes. (**b**) The number of genomes potentially capable of de novo synthesizing IAA from Trp. (**c**) The number of genomes potentially competent to synthesize IAA from intermediates. The density plots (**d**–**f**) show the distribution of the number of genomes possessing coexisting pathways. The *x* axes represent the number of coexisting disparate synthesis pathways in an individual genome and *y* axes represent the density of the number of genomes.

**Figure 3 molecules-24-01411-f003:**
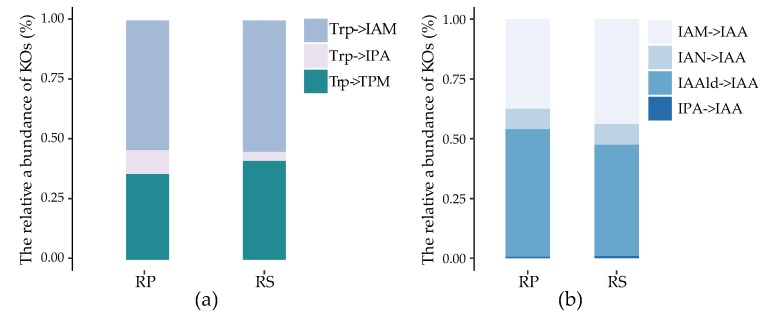
Quantification of the IAA synthesis related genes in root environments. The bar plots show the relative abundance of (**a**) Trp-catalyzing enzymes and (**b**) enzymes in the last step of IAA synthesis in rhizosphere (RS) and rhizoplane (RP), respectively. (**c**) and (**d**) represent the taxonomic structures of Trp-catalyzing enzymes and IAA synthesis enzymes, respectively.

**Figure 4 molecules-24-01411-f004:**
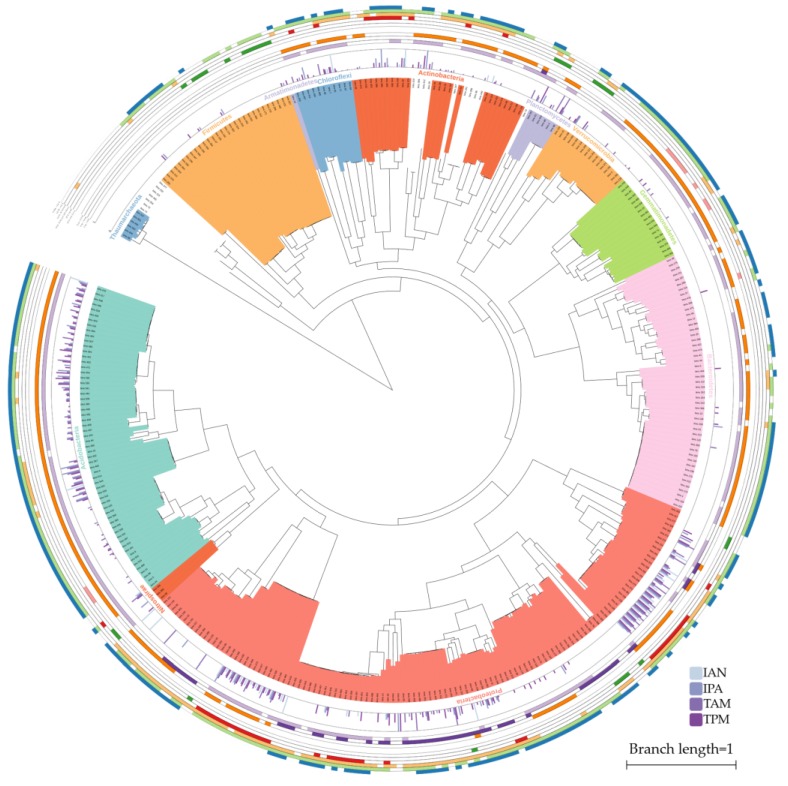
The phylogeny and distribution of IAA synthesis pathways in rhizobacteria. The tips of the tree are masked with different colors to indicate corresponding phylum. The histograms represent the percent of genes in each pathway located on plasmids. The three inner rings and six outside rings show the distribution of different Trp-catalyzing enzymes and capacities for synthesizing IAA from intermediates by different pathways, respectively. The blank space along the rings represent that no related genes or pathways are found in the rhizobacterium.

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
