# Peer review of "The Distribution of Tryptophan-Dependent Indole-3-Acetic Acid Synthesis Pathways in Bacteria Unraveled by Large-Scale Genomic Analysis"

_molecules, 2019, doi:10.3390/molecules24071411_

Round 1

Reviewer 1 Report

The authors studied bacteria community living in plant roots and they found 82% of bacterial genomes have IAA synthesizing pathways. It’s quite an interesting study, and IAA is important for the growth of plants. There are only minor points as below.

1. First use of a pathway (IAM, line 24), a genus name (A. tumefaciens, line 40) should be spelled out.

2. Italic font for some specific terms: gene names (vir regulon, chv genes – line 40; iaaM, iaaH – line 48-49, 219); bacterial name (Pseudomonas fluorescencs – line 58); de novo (line 110, 123, 286, 297); the whole line 202

3. Error font: Line 308-309, 332-334; Reference list.

4. Grammar checks: various stresses (line 41), past tense (line 76), comma (line 116), space (256)

5. It’d be more convinced to add citation (line 44, line 77)

6. I think it’d be better to break down the sentence from line 40 to 42, since it’s too long and difficult to understand. From line 66-69, they lack of connections between sentences. Line 134, the sentence is unclear, the percentage in particular. From line 154-155, it’d be easier to understand if they add more detail, like which single complete pathway they want mention. 

7. The results should be summarized. There are also many numbers, percentages for each phylum and it’s confusing. I think a summarized table is needed. 

8. Supplementary table 2: row 17 grammar checks – “Synthesis”

9. Supplementary table 3: What is IPV? The term isn’t mentioned in the manuscript.

Author Response

Response to Reviewer 1 Comments

Thank you for your useful comments and suggestions on our manuscriptwe have modified the manuscript accordingly.

Point 1.   First use of a pathway (IAM, line 24), a genus name (A. tumefaciens, line 40) should be spelled out.

Response 1: We have spelled out the full name of the terms ‘indole-3-acetamide (IAM)’ (line 24) and ‘Agrobacterium. tumefaciens’ (line 40).

Point 2.   Italic font for some specific terms: gene names (vir regulon, chv genes – line 40; iaaM, iaaH – line 48-49, 219); bacterial name (Pseudomonas fluorescencs – line 58); de novo (line 110, 123, 286, 297); the whole line 202.

Response 2: We have corrected the fonts of all these words.

Point 3.   Error font: Line 308-309, 332-334; Reference list.

Response 3: We have modified the font ‘Times’ into ‘Palatino Linotype’ for these lines (line 311-313, and line 335-338).

Point 4.   Grammar checks: various stresses (line 41), past tense (line 76), comma (line 116), space (256).

Response 4: We have corrected these grammar errors by replacing ‘various stress’ with ‘various stresses’ (line 42), ‘is’ with ‘was’ (line 77) and adding a comma between ‘30.8% Actinobacteria, 15.3% Bacteroidetes’ (line 117) and deleting a space among the words ‘a recent study’ (line 259).

Point 5.   It’d be more convinced to add citation (line 44, line 77)

Response 5: We have added citations to these statements (line 44, line 78).

Point 6.   I think it’d be better to break down the sentence from line 40 to 42, since it’s too long and difficult to understand. From line 66-69, they lack of connections between sentences. Line 134, the sentence is unclear, the percentage in particular. From line 154-155, it’d be easier to understand if they add more detail, like which single complete pathway they want mention.

Response 6: We have re-written sentences from line 40 to 42 and sentences from line 66 to 69 and line 134 to make them clearer and easy-to-follow. We have added details to the results from line 154 to 155.

Original line 40-42: the addition of IAA represses the vir regulon and chv genes of A tumefaciens, and overproduction of IAA in Sinorhizobium meliloti RD64 exhibits resistance to various stresses, such as UV and salinity; and also activates the tricarboxylic acid (TCA) cycle.

Revised line 40-42: The addition of IAA represses the vir regulon and chv genes of Agrobacterium tumefaciens. Overproduction of IAA in Sinorhizobium meliloti RD64 exhibits resistance to various stresses, such as UV and salinity; and also activates the tricarboxylic acid (TCA) cycle.

Original line 66-71: It is reported that nitrilase-based and NHase/amidase system-based synthesis pathways have very different regulatory mechanisms. P. agglomerans is a gall-inducing bacterium and possesses IPA and IAM pathways. It’s reported that the IPA pathway is related to plant colonization and the IAM pathway is associated with pathogenicity.

Revised line 66-71: It is reported that nitrilase-based and NHase/amidase system-based synthesis pathways have very different regulatory mechanisms. What’s more, different pathways function under different physiological status. Pantoea. agglomerans is a gall-inducing bacterium and possesses IPA and IAM pathways. It’s reported that the IPA pathway is related to plant colonization and the IAM pathway is associated with pathogenicity.

Original line 136-138: the one-step nitrilase pathway was preferred by most bacteria (56.8%). However, only a minority of bacterial nitrilases were reported to be capable of converting IAN into IAA.

Revised line 136-138: 56.8% of bacterial genomes possess the the one-step nitrilase pathway. However, only a minority of bacterial nitrilases identified by in silico analysis were experimentally reported to be capable of converting IAN into IAA.

Original line 156-158: we observed that 5 genomes from Proteobacteria possessed 3 different intact IAA synthesis pathways (Figure 2e), but a plethora of genomes exclusively owned one complete IAA synthesis pathway.

Revised line 156-158: we observed that 5 genomes from Proteobacteria possessed 3 different intact IAA synthesis path-ways, both of which simultaneously possess IPA, IAM and TPM pathways (Figure 2e), but a plethora of genomes exclusively owned one complete IAM or TPM synthesis pathway.

Point 7.   The results should be summarized. There are also many numbers, percentages for each phylum and it’s confusing. I think a summarized table is needed.

Response 7: The supplementary tables from 1 to 4 have summarized all the numbers for each phylum and each pathway for each part of the analysis. To make the tables clearer, we have added titles to the tables.

Point 8.   Supplementary table 2: row 17 grammar checks – “Synthesis”.

Response 8: We have corrected the word ‘Synthesisi’ to ‘Synthesis’.

Point 9.   Supplementary table 3: What is IPV? The term isn’t mentioned in the manuscript.

Response 9: ‘IPV’ is a spelling mistake for ‘IPA’ pathway as mentioned in the article, as some papers use ‘IPV’ for the abbreviation for ‘indole-3-pyruvate’ rather than IPA. We have corrected this mistake in Supplementary table 3.

Reviewer 2 Report

The manuscript "The distribution of tryptophan-dependent indole-3-acetic acid synthesis pathways in bacteria unraveled by large-scale genomic analysis" by Zhang et al. deals with the presence/absence and distribution of IAA synthesis pathways among several bacterial phyla using a large-scale genomic analysis. Moreover, the authors quantified the IAA synthetic genes in root environments, namely the rhizosphere and rhizoplane of Setaria italica roots.

I find the manuscript well written and well-presented and makes a novel contribution to our understanding of the complex IAA synthesis systems in bacteria combining large-scale genomic analysis of available bacterial genomes and their presence, role/importance on plant roots.

I only have some minor revisions regarding text formatting that the authors should consider to revise in order to match the journal's requirements, in particularly expressions that should be always italicized, like "de novo" and the list of references.

Please find in attach the detail comments of the manuscript  molecules-478577.

Author Response

Response to Reviewer 2 Comments

Thank you for your elaborative review on our manuscript. You have made great summary to our paper and given us promising suggestions for further analysis. We have modified the manuscript accordingly.

Point 1: Line 40 please add complete name for Agrobacterium tumefaciens.

Response 1: We have spelled out the full name of the bacterial name (line 40).

Point 2: Line 58 "Pseudomonas fluorescens" should be in italics

Response 2: We have corrected the font of the words (line 58).

Point 3: Line 67 please add complete name for Pantoea agglomerans.

Response 3: We have spelled out the full name of the bacterial name (line 68).

Point 4: Lines 96-97 please replace semicolons by end point.

Response 4: We have replaced the writing error of the punctuation ‘;’ with ‘.’ (line 98).

Point 5: Line 100 - "in vitro" should be in italics.

Response 5: We have modified the font into ‘Italics’ (line 101).

Point 6: Figure 2- please correct the colors in the figure caption to match with the ones in the figure

Response 6: We have replaced the Figure 2 with a new one that have the correct colors in the figure legend matching the colors in the figure.

Point 7: Line 123 - "de novo" should be in italics. check throughout the text

Response 7: We have corrected the font error throughout the whole text.

Point 8: Line 204 and throughout the text - if possible replace ">= or <= " by ≥ or ≤.

Response 8: We have replaced the symbols ‘>= or <=’ with the suggested one ‘ or ≤’ throughout the whole text.

Point 9: Throughout the text- Add a space before each reference within brackets.

Response 9: We have added a space before each reference.

Point 10: Line 402-559 - Please reformat References in the list according to the journal requirements.

Response 10: We have modified the format of reference list to accord with the journal requirements (line 408-550).